# SYNTHETIC DATASETS FOR NEURAL PROGRAM SYNTHESIS

**Richard Shin**[*]
UC Berkeley

**Neel Kant**
UC Berkeley and ML@B[†]

**Kavi Gupta**
UC Berkeley

**Christopher Bender**
UC Berkeley and ML@B

**Brandon Trabucco**
UC Berkeley and ML@B

**Rishabh Singh**
Google Brain

**Dawn Song**
UC Berkeley

## ABSTRACT

The goal of program synthesis is to automatically generate programs in a particular language from corresponding specifications, e.g. input-output behavior. Many current approaches achieve impressive results after training on randomly generated I/O examples in limited domain-specific languages (DSLs), as with string transformations in RobustFill. However, we empirically discover that applying test input generation techniques for languages with control flow and rich input space causes deep networks to generalize poorly to certain data distributions; to correct this, we propose a new methodology for controlling and evaluating the bias of synthetic data distributions over both programs and specifications. We demonstrate, using the Karel DSL and a small Calculator DSL, that training deep networks on these distributions leads to improved cross-distribution generalization performance.

## 1 INTRODUCTION

Program synthesis is one of the central problems in Artificial Intelligence studied from the early days (Manna & Waldinger, 1971; Waldinger & Lee, 1969) and has seen a lot of recent interest in the machine learning and programming languages community (Alur et al., 2013; Gulwani et al., 2012; 2015; Muggleton, 1991; Lin et al., 2014; Solar-Lezama, 2013). The recent neural approaches can be broadly classified into two categories: *program induction* and *program synthesis*. Both approaches share the objective of learning program semantics but do so in different ways. Program induction aims to embed the semantics of a particular algorithm into a differentiable model trained end-to-end, whereas the goal of program synthesis is for a model to learn the semantics of a domain-specific language (DSL) and produce programs defined by corresponding specifications. Both problems necessitate large datasets, either of I/O pairs in the case of program induction, or programs with corresponding I/O pairs in the case of program synthesis.

However, since large datasets for program induction and synthesis tasks do not exist, these approaches train models on large synthetically generated datasets. Presumably, if a model can accurately predict *arbitrary* program outputs (for induction) or programs in the DSL (for synthesis) then it has likely learnt the correct algorithm or DSL semantics.

Although this approach has led to some impressive synthesis results in many domains, synthetically generating datasets that cover all DSL programs and the corresponding input space can be problematic, especially for more complex DSLs like *Karel the Robot* (Pattis, 1981) which includes complex control-flow primitives (while loops and if conditionals) and operators. Likewise, for induction tasks, the sampling procedure for program specifications may lead to undesirable biases in the training distribution that inhibit strong generalization.

In this paper, we consider two problem settings. The first is the Karel domain and the recently proposed Karel synthesis model (Bunel et al., 2018). We identify many distributions of input examples and DSL programs for which the Karel synthesis model performs poorly. The second problem setting is a program induction problem in which a model is trained to execute and predict the output of simple arithmetic expressions, which we denote the *Calculator* domain; for this domain,

---

[*]`ricshin@cs.berkeley.edu`
[†]Machine Learning at Berkeley

we considered common synthetic data generation strategies including one from `tensor2tensor` (Vaswani et al., 2018), an open-source deep learning library. Upon analysis, we find evidence of undesirable artifacts resulting from certain biases in the generation algorithm.

Our results indicate that models trained with common methodologies for synthesizing datasets fail to learn the full semantics of the DSL, even when they perform well on a test set, and suggest the need of a more principled way to generate synthetic datasets. For some program and input distributions, the state-of-the-art neural synthesis models perform quite poorly, often achieving less than $5\%$ generalization accuracy. In our paper, we develop a new methodology for creating training distributions over programs in the DSL to mitigate some of these issues. Moreover, unlike previous works that have ignored considering the distributions over input space, we show that input distributions also play a significant role in determining the synthesizer performance. Our methodology involves defining the distribution over DSL programs and input space using a set of random variables to encode much of the valuable features which describe the data, e.g. in the Karel domain, the amount of control flow nesting in programs or the number of markers present in the inputs.

Our methodology allows us to identify several specialized distributions over the input space and Karel programs on which the current state of the art synthesis models (Bunel et al., 2018) perform poorly when trained on traditional program and input distributions, and tested on our new distributions. From this, we design new training distributions by ensuring greater uniformity over the random variables in our methodology. By retraining the same architecture on these new training data distributions, we observe a greater ability to generalize, with significant improvements when evaluated on the aforementioned test sets. We also observe similar improvements in the *Calculator* domain as well.

This paper makes the following key contributions:

- We propose a new methodology to generate different desirable distributions over the space of datasets for program induction and synthesis tasks.
- We instantiate the methodology for the Karel and Calculator domains and show that model generalization is worse on datasets generated by our technique.
- We then retrain models in both domains and demonstrate that models achieve greater overall generalization performance when trained on datasets generated with our methodology.

## 2 RELATED WORK

### 2.1 TRAINING MODELS WITH SYNTHETIC DATA

In certain domains like computer vision and robotics, collecting high-quality real-world training data incurs significant cost, and so many researchers have investigated the use of large amounts of synthetic data. For example, Christiano et al. (2016); Peng et al. (2017); Pinto et al. (2017); Bousmalis et al. (2017) aim to learn robotics policies that compensate for differences between the real world and the simulation. Within computer vision, Shrivastava et al. (2016) demonstrate learning from entirely synthetic images for gaze and pose estimation.

### 2.2 NEURAL PROGRAM INDUCTION AND SYNTHESIS.

Program induction methods learn differentiable modules such as stack (Joulin & Mikolov, 2015), RAM (Kurach et al., 2016), GPU (Kaiser & Sutskever, 2015), and read-write external memory (Graves et al., 2014) to represent algorithms. Other methods attempt to learn differentiable control flow operations (Gaunt et al., 2016; Neelakantan et al., 2015). These approaches reconstruct outputs given inputs, inferring the underlying algorithms. Other methods instead learn neural modules from program traces rather than from I/O examples (Reed & de Freitas, 2015; Xiao et al., 2018; Cai et al., 2017).

Devlin et al. (2017); Parisotto et al. (2017) use neural program synthesis techniques for learning string editing programs in RobustFill. Similarly, Balog et al. (2016) learn array programs in DeepCoder, and Bhupatiraju et al. (2017) learn to compose API calls. Bunel et al. (2018) apply neural program synthesis to the Karel domain we consider in our paper; we use their architecture and dataset. Many of these approaches report high test performance, but good performance on synthetically-specified programs may not indicate the model's ability to generalize to arbitrary user-desired programs. In

fact, our results show that under certain distributions, these models perform quite poorly. To verify these models appropriately generalize to reasonably complex arbitrary programs, the test set should sufficiently represent the universe of these programs and their specifications. Likewise, to avoid biasing the result, the test set should also draw uniformly from these programs and specifications. For example, for RobustFill, the data generation methodology only sampled programs uniformly from the string DSL, but did not take into account the distribution over input strings such as their length, frequencies of occurrence of regular expressions and their nesting, common words and constants etc.

## 3    DATA GENERATION METHODOLOGY

Currently, automated data generation focuses in large part on a constructive process, whose parameters can be tuned. We propose a complementary approach in which we perform a subsequent filtering step on this process to ensure that the resulting distribution has certain properties.

We define a *salient* random variable as one whose distribution in the final dataset is of interest. In the case of program synthesis, we consider two kinds of salient variables: variables denoting important features of a program in the given DSL, such as its length and degree of nesting; and variables denoting features for the input space.

In many cases, we can modify our sampling procedure to ensure a desirable distribution of a particular salient variable. However, for some salient variables, it is infeasible to tune the parameters of a given sampling procedure in order to obtain a desired distribution for that salient variable. For example, if we sample programs directly from a context-free grammar, it is difficult to control the distribution of various salient variables such as program length, degree of nesting, etc. This is a notable problem in both the Karel and Calculator domains.

Furthermore, within the context of program synthesis specifically, there is often an additional challenge: not all inputs are valid for all programs. For example, in the Karel domain, an input for a given program would be invalid if the program attempts to perform illegal actions for that input (such as `move` into walls or `pickMarker` in a cell containing no markers). The requirement that the program/input pairs suit each other itself acts as an unpredictable filter that makes it difficult to ensure uniformity of salient variables by tuning generation parameters.

As such, we propose a methodology for randomly sampling a dataset $\mathcal{D}$ (consisting of elements of $\mathbb{S}$) while ensuring that a given salient variable $\nu : \mathbb{S} \to \mathbb{X}$ (where $\mathbb{X}$ is finite and discrete), denoted as a random variable $X = \nu(s)$, has a uniform distribution throughout $\mathcal{D}$. To do this, we first sample an example $s \sim q(\cdot)$ from an original distribution $q$. We then add $s$ to $\mathcal{D}$ with probability $g(s)$, where

$$g(s) = (P_q[X = \nu(s)] + \varepsilon)^{-1} \left( \min_{x \in \mathbb{X}} P_q[X = x] + \varepsilon \right),$$

with $P_q[X]$, the probabilities induced over $X$ via $q$, calculated empirically based on counts computed with past samples drawn from $q$. We repeat the above until $\mathcal{D}$ is of a desired size. For full pseudocode see Section B.1 in the appendix.

We use $\varepsilon \in \mathbb{R}^+$ as a hyperparameter to trade off the runtime of the above procedure with the level of $X$'s uniformity in $\mathcal{D}$. In Section B.2 (in the Appendix), we provide a probabilistic bound on the uniformity of $X$ in the resulting distribution, for the case where $\varepsilon = 0$. Also, for when $\varepsilon > 0$, we can show that drawing a single sample is possible in $O(\frac{1}{\varepsilon})$ calls to the original sampler (proof in Section B.3, with empirical experiments in Section B.4 and B.5). Increasing $\varepsilon$ increases the algorithm's speed, at the cost of allowing the distribution of $X$ in $\mathcal{D}$ to further diverge from uniform.

## 4    APPLICATION TO KAREL: EXPERIMENTS WITH NEW TEST DISTRIBUTIONS

Karel is an educational programming language (Pattis, 1981) where the programmer writes imperative programs with conditionals and loops to produce a sequence of actions for an agent (a robot named Karel) which lives in a rectangular $m \times n$ grid world. For a detailed description of the particular instantiation of the Karel language and input grid specification that we consider, see Section A. The program synthesis task that we consider is as follows: given a set of pairs of input and output grids $\{(i_1, o_1), \cdots, (i_n, o_n)\}$, find a Karel program $\pi$ such that executing $\pi$ on $i_1$ results in $o_1$, $i_2$ results

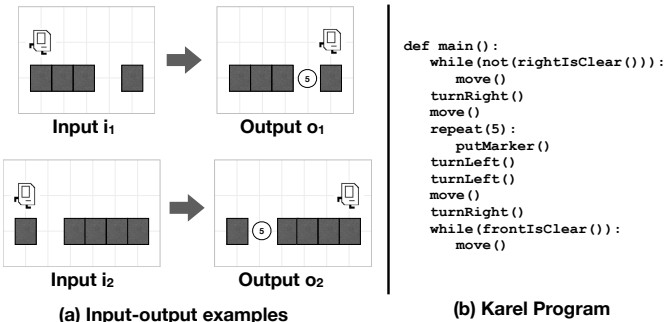

Figure 1: An example Karel synthesis task, where the goal is to synthesize the program in (b) given the two I/O examples shown in (a). See the Appendix for more details about Karel.

in $o_2$, and so on. In the paper, $n = 5$ unless otherwise specified. An example Karel synthesis task with the I/O examples and the corresponding Karel program to be synthesized is shown in Figure 1.

In this section, we employ the Karel instantiation of our abstract data generation methodology in Section 4.1 to generate different test datasets. By imposing a more uniform distribution over the salient random variables when generating the I/O specifications and target programs which make up the test set, we observe much lower accuracies of the previous Karel synthesis models (Bunel et al., 2018) compared to the original test set.

## 4.1 SALIENT VARIABLES IN KAREL

We devised the following salient random variables to describe the input space in Karel:

- *Grid size*: Dimensions of the grid in which Karel can act.
- *Marker ratio*: Fraction of cells with at least one marker.
- *Wall ratio*: Fraction of cells which contain a wall.
- *Marker count*: Number of markers that are present in a cell containing markers.
- *Number of grids*: Number of I/O grids shown to the model to specify the desired program.

For the program space in the Karel DSL, we consider the following random variables:

- *Program size*: Size of the program in terms of number of tokens.
- *Control flow ratio*: Number of control flow structures appearing in the program.
- *Nested control flow*: The amount of control flow nesting in programs (e.g. while inside if).

## 4.2 CHANGING THE I/O DISTRIBUTION

We reproduced the encoder-decoder model of Bunel et al. (2018) and trained it using the provided synthetic training set with the teacher-forcing maximum likelihood objective. On the existing test set, our model achieves 73.52% generalization accuracy, slightly higher than the 71.91% accuracy reported in Bunel et al. (2018). *Generalization accuracy* denotes how often the model's output is correct on both the 5 I/O examples shown to the model and the remaining held-out 6th I/O example.

To test how the model may be sensitive to changes in the I/O examples used to specify the program, we created new test sets by sampling new input grids and running them on each of the programs in the existing test set to obtain new I/O pairs. By keeping the programs themselves the same, we avoid inadvertent changes in the inherent difficulty of the task (the complexity of the programs to be synthesized).

**Salient random variables with uniform distribution.** We first generated grids such that they would follow a distribution that is as uniform as possible in the salient features in Section 4.1. We used the following procedure to sample each grid: 1) sample the *grid size* (height and width) from

Table 1: Generalization accuracies of the baseline model and a model trained on a uniform I/O distribution, on selected datasets. $G, U$, and $A$ stand for $Geom(0.5), \mathcal{U}\{1, \dots 9\}$ and $10 - Geom(0.5)$ respectively. See Section 4.2 for dataset generation details, and Section 5.1 for details about the Uniform model.

| $r_{\text{wall}}$ | 0.05 | | | 0.25 | | | 0.65 | | | 0.85 | | |
| $r_{\text{marker}}$ | 0.85 | | | 0.65 | | | 0.25 | | | 0.05 | | |
| $\mathcal{D}_{\text{marker count}}$ | $G$ | $U$ | $A$ | $G$ | $U$ | $A$ | $G$ | $U$ | $A$ | $G$ | $U$ | $A$ |
|---|---|---|---|---|---|---|---|---|---|---|---|---|
| Baseline (%) | 24.30 | 1.32 | 0.04 | 21.08 | 2.98 | 0.08 | 16.63 | 13.31 | 6.63 | 15.99 | 12.88 | 12.98 |
| Uniform (%) | 69.37 | 70.21 | 68.99 | 63.25 | 63.74 | 62.78 | 65.83 | 67.39 | 68.09 | 77.32 | 78.63 | 80.19 |
| $\Delta$ | +45.07 | +68.89 | +68.95 | +42.17 | +60.76 | +62.70 | +49.20 | +54.08 | +61.46 | +61.33 | +65.75 | +67.21 |

$x, y \sim \mathcal{U}\{2, \dots 16\}$; 2) sample the *marker ratio* $r_{\text{marker}} \sim \mathcal{U}(0, 1)$ and *wall ratio* $r_{\text{wall}} \sim \mathcal{U}(0, 1)$; 3) for each cell $(i, j), 0 \leq i < x, 0 \leq j < y$ in the grid, sample $m_{i,j} \sim Bernoulli(r_{\text{marker}})$ and $w_{i,j} \sim Bernoulli(r_{\text{wall}})$; 4) if $m_{i,j} = 1$ and $w_{i,j} = 0$, sample *marker count* $mc_{i,j} \sim \mathcal{U}\{1, \dots 9\}$, otherwise set $mc_{i,j} = 0$; 5) place walls and markers in grid according to $w_{i,j}$ and $mc_{i,j}$; 6) place Karel at a random location (not containing a wall) and with a random orientation. After generating 5 input grids for a given program, we ensure that the program does not crash on any of them and also check whether the 5 input grids exhibit complete *branch coverage* (i.e., each branch is taken by at least one of the 5 inputs). If either of these conditions are not satisfied, we discard all 5 grids and start over.

On this dataset, the model trained on existing data achieved generalization accuracy of only 27.9%, which was a drop of 44.6pp from the existing test set's generalization accuracy of 73.52%.

**Salient random variables with narrow distributions.** We further investigated the performance drop noted above by synthesizing "narrower" datasets that captured different parts of the joint probability space over the salient input random variables. For each narrow dataset, we selected $r_{\text{wall}}$ and $r_{\text{marker}}$ (both between 0 and 1) as well as a distribution $\mathcal{D}_{\text{marker count}}$ which would be the same for all I/O grids. Then, we follow the procedure below for each grid: 1) sample the *grid size* (height and width) $x, y \sim \mathcal{U}\{10, \dots 16\}$; 2) randomly choose $xy \cdot r_{\text{wall}}$ cells to contain walls, and $xy \cdot r_{\text{marker}}$ cells for markers; 3) sample $mc_{i,j} \sim \mathcal{D}_{\text{marker count}}$ for all cells $(i, j)$ chosen to contain markers; 4) place Karel at a random location (not containing a wall) and with a random orientation. In our experiments, we primarily used 3 different distributions for $\mathcal{D}_{\text{marker count}}$: $Geom(0.5)$ truncated at 9, $\mathcal{U}\{1, \dots 9\}$, and $10 - Geom(0.5)$ which, when sampled, has a value equal to 10 minus a sample from Geom(0.5), truncated at 1.

The results are shown in Table 1 (row 1, "Baseline (%)"). We discovered that the most correlated factor with model performance was the distribution $\mathcal{D}_{\text{marker count}}$. A more negative skew consistently lowered model performance, and this effect was more pronounced at higher values of $r_{\text{marker}}$.

### 4.3 CHANGING THE PROGRAM DISTRIBUTION

We will now examine how the existing model can surprisingly fail to perform well at synthesizing certain programs that are different from those in the existing validation and test sets.

**Performance on complex DSL constructs.** We examined whether or not the model could succeed in synthesizing programs which require nesting of conditional constructs. This was of interest since these programs were relatively rare in the training dataset. We generated an evaluation dataset comprised solely of programs that contained `while` inside `while` statements, and another dataset in which all programs had `while` inside `if` statements.[1] We found that the model fared very poorly on these datasets, achieving only 0.64% and 2.23% accuracy respectively.

**Programs only containing actions.** Intuitively, much of the difficulty in the Karel program synthesis task should come from inferring the control flow statements, i.e. `if`, `ifElse`, and `while`. Synthesizing a Karel program that only contains actions is intrinsically a much more straightforward task, which a relatively simple search algorithm (such as A[*]) can perform well.

---

[1]To avoid any negative effects from changes in the I/O distribution, we attempted to ensure that $r_{\text{wall}}, r_{\text{marker}}$ and $\mathcal{D}_{\text{marker count}}$ matches that of the provided training and test sets.

Table 2: Results on programs only containing actions. The generalization accuracy on action-only programs in the existing test set is 99.24%. See Section 5 for details on Action-Only Augmented.

| Model type | Program length | | | | | | | |
|---|---|---|---|---|---|---|---|---|
| | 1 | 2 | 3 | 4 | 5 | 6 | 7 | 8 |
| Baseline | 16.00% | 30.00% | 44.24% | 52.88% | 56.56% | 66.94% | 67.16% | 73.06% |
| Action-Only Augmented | 20.00% | 41.60% | 52.24% | 61.72% | 63.04% | 72.20% | 72.74% | 78.12% |

We performed an experiment using test datasets generated by enumerating action-only programs of various lengths. As there are five actions (`move`, `turnLeft`, `turnRight`, `putMarker`, `pickMarker`), there exist $5^L$ textually unique action-only programs for length $L$. We sampled up to 500 unique programs of lengths $1, 2, \ldots, 8$. For each program, we generated 10 specifications, each containing 5 I/O pairs. We sampled each I/O pair from the set of all input grids in the existing training data (of which there are 6.7 million), as to match its distribution as closely as possible.

Table 2 shows the results. Remarkably, even though the underlying programs have relatively low complexity, the model's accuracy is lower on every one of these action-only test sets than the existing provided test set. The generalization accuracy grows as the program length becomes longer, even though those programs should be harder to synthesize.

Among the existing action-only programs in the test set, the model's generalization accuracy on that subset is 99.24%. Given the surprising nature of this result, we investigated the difference between the action-only programs we generated, and those in the existing test set. We found that in the existing training and test sets, all programs contain at least two actions, and also contain at least one `move` action somewhere in the program. These and any undiscovered differences in the distribution of programs seem to have caused the gap in performance.

## 5 APPLICATION TO KAREL: CHANGING TRAINING DISTRIBUTIONS

In Section 4, we saw that the existing model performs much more poorly on certain test datasets that we constructed, compared to its performance on the existing test set as reported in Section 4.2. In light of the framework in Section 3, various imbalances of the salient random variables in the existing training data could have caused these gaps in performance. Then, a natural solution is to train using datasets constructed to avoid undesirable skews in the salient random variables, which should hopefully perform better across a variety of distributions.

### 5.1 TRAINING DATASETS WITH UNIFORM I/O

We generated a training dataset by taking the programs of the existing training set and synthesizing I/O pairs using the procedure described in Section 4.2. Furthermore, to make the "number of grids" salient variable uniform, we modify the training procedure by uniformly sampling a number between 1 and 5 for each mini-batch, and using that many I/O grids to specify the program to the model. We trained a model on this data and then evaluated it on the same set of narrow distribution evaluation datasets as mentioned in Section 4.2. Table 1 and Figure 2 compares how this new model performs to the baseline model. The model trained on uniform I/O distributions maintains much higher generalization accuracy on the test sets of Section 4.2 than the baseline model. Note that the uniform I/O distribution is not simply a union of the tested distributions and is intended to cover all possible input specifications.

### 5.2 REAL-WORLD BENCHMARKS

We evaluated both the baseline model, and the uniform model described in the previous paragraph, on a set of 36 real-world Karel programming problems. This dataset was compiled from the Hour of Code Initiative and Stanford University's introductory computer science course, CS106A, with the problems being hand-designed as educational exercises for students. We found that the baseline model got 4 correct (11.1%) while the uniform model got 7 correct (19.4%) when both models were trained with 5 shown I/O pairs, i.e., without making the "number of grids" salient variable uniform.

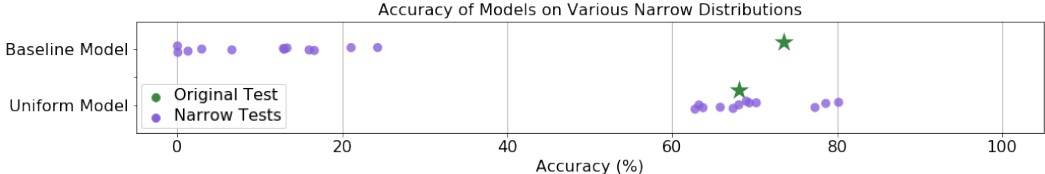

Figure 2: Comparison of generalization accuracies across the datasets given in Table 1 plus performance on the original test set. The training datasets used for both models (denoted *Baseline* and *Uniform*) contained the same programs; however, for *Uniform*, we sampled new I/O grids used to specify the programs, with homogenized salient random variables, as described in Section 4.2.

This further demonstrates the uniform model's increased ability to generalize to out-of-distribution datasets, including those which are of interest to humans. Both models' accuracies are still low compared to the performance on the synthetically generated test set.

After further analysis, we believe the models face two challenges on the real-world test set: (1) many of the real-world problems require long programs to solve compared to the synthetic test set, (2) the specifications in the real-world examples always contains fewer than 5 I/O pairs, and often only a single one. However, the original training methodology for the model assumes that it is provided with a diverse set of 5 I/O pairs. When we modified the training procedure to vary the number of shown I/O examples as in Section 5.1, the baseline model got 12 correct (33.3%) and the uniform model got 11 correct (30.6%). This shows that the homogenization on the number of I/O pairs was effective. Overall, the real-world dataset's I/O distribution was similar to the MSR training datset in terms of the salient random variables we homogenized, so it is unsurprising that the baseline model was able to outperform the uniform model, consistent with the results on the MSR test set in Figure 2.

### 5.3 Augmenting Dataset with Action-only Programs

We observed in Section 4.3 that the model fails to do well on either action-only programs or programs with many control-flow statements. In the case of action-only programs, we found that the training data had been pruned to only include programs with at least two actions and at least one `move`, and in the case of programs with complex control flow, we found a similar sparsity in the train set.

As discussed in Section 3, the principled way to counteract this sparsity is to introduce uniformity into a set of salient variables. This methodology allows us to counteract both naturally sparse data (such as complex control-flow) and spurious data pre-processing (such as enforcing programs to have at least two actions).

In our case, we introduce uniformity into the length of action-only programs by synthesizing 20,000 programs of each length 1 to 20, by uniformly selecting tokens from the five action choices and generating I/O with other salient random variables as close to the original training set as possible; we append these new programs to the original dataset to train a new model. Table 2 shows a clear improvement when homogenizing this salient random variable. The new model achieved 71.8% accuracy on the original test set, which is very close to the 73.52% accuracy of the baseline model.

### 5.4 Training Datasets with Narrowly Distributed I/O

As done in for the uniform training dataset, we generated "narrow" training datasets by keeping the same programs as in the existing training data and replacing the I/O pairs with the process from Section 4.2.

We trained a variety of models and evaluated them on 12 datasets of different I/O feature distributions. Figure 3 summarizes the results of evaluating each model on each dataset by noting the performance of the model on the narrow dataset of the same type and the outcome on every other narrow dataset. For the models trained on the low variance datasets, we observed that they all consistently achieved between 60 and 70 percent accuracy on their own training distribution; however, the uniform model was able to achieve similar performance (between 57 and 80 percent accuracy) as shown in Table 1 and Figure 2. As such, we hypothesize that models trained on wide, uniform distributions can still

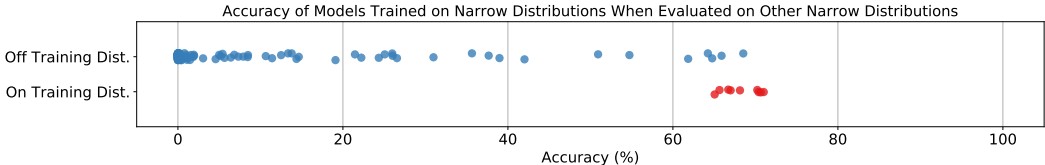

Figure 3: Evaluation of models that were trained on narrow distributions. When evaluated on their own training distribution, they consistently achieved similar results. When evaluated on a different narrow distribution, the performance was rarely similar and usually very low.

perform comparable to models trained on narrow sub-distributions, even when tested on the same sub-distributions.

## 6 APPLICATION TO CALCULATOR

The Calculator task is given as follows: given an expression such as `"5+4*(2+3)"`, compute the result modulo 10; in this case, 5. Calculator is a program induction task rather than a program synthesis task like Karel; nevertheless, creating data for the Calculator problem involves sampling from a context-free grammar. Additionally, Calculator is not as intricate a domain as Karel and thus we can more completely control the environment of data generation with less fear of lurking variables.

### 6.1 CALCULATOR ENVIRONMENT

**Calculator model.** Similar to the work by Zaremba & Sutskever (2014), we implement an LSTM that parses calculator expressions on a character level. We perform a 10-class classification problem using a dense network on the final hidden state of the LSTM. The prediction is correct if it exactly matches the evaluation result of the expression, modulo 10.

**Distributions of Calculator tasks.** We propose 4 distributions for calculator tasks: direct CFG sampling (`DCFG`), tensor2tensor sampling (`T2T`), "runs" CFG sampling (`RCFG`), and balanced sampling (`BAL`).

Two of our distributions represent reasonable ways in which a researcher might choose to sample data. The first is `DCFG`, which involves returning a digit with some probability $(1 - p)$, or else recursively sampling two productions and combining them with a $-$, $*$, or $+$, each with probability $\frac{p}{3}$. This corresponds to a direct, weighted sampling of the CFG for the calculator grammar. The second is `T2T`, which is used by the tensor2tensor library to sample arithmetic expressions in one of its examples.[2] It involves sampling a depth $d$, then ensuring that the resulting AST has depth $d$ by forcing a random side of the operation production to be sampled to $d - 1$ and the other side to be sampled to a depth $d' \sim \mathcal{U}\{0, 1, \ldots, d - 1\}$.

The other two distributions represent potentially difficult or nonstandard problems that might appear in practical environments. `RCFG` is similar to `DCFG` but involves increasing the frequency of "runs" of the associative operations $+$ and $*$ by picking 2, 3, or 4 subexpressions and then combining them with the given symbol. `BAL` (balanced sampling) involves selecting a depth and then creating an AST that is a balanced binary tree at that depth. Importantly, regardless of sampling technique, redundant parentheses are removed. This is to increase the difficulty somewhat as order of operations needs to be established.

### 6.2 SALIENT VARIABLES AND METHODOLOGY

We use the following salient variables: length (rounded to the nearest even number), number of operations, number of pairs of parentheses, mean parenthesized depth, and maximum parenthesized

---

[2]`https://github.com/tensorflow/tensor2tensor/blob/8bd81e8fe9dafd4eb1dfa519255bcbe3e33c7ffa/tensor2tensor/data_generators/algorithmic_math.py`

|  | Original | Length | Max Depth | Mean Depth | #Operations | #Parens |
|---|---|---|---|---|---|---|
| T2T | 83.83% | +4.35pp | +4.24pp | +2.14pp | +1.19pp | +2.32pp |
| DCFG | 78.25% | +3.84pp | +5.92pp | +4.02pp | +6.72pp | +4.51pp |

Table 3: Improvements in Calculator performance over unhomogenized distributions when various homogenizations were applied. See Section 6 for details on performance metrics.

depth. Parenthesized depth is defined for each digit and refers to the number of nested parentheses it is in. For example in `(1+2)*(3-4)+5`, the 1, 2, 3, and 4 are at depth 1 while the 5 is at depth 0.

We constructed $2 \times (1 + 5)$ distributions in total, corresponding to a total of 2 task distributions, T2T and DCFG, which represent the "natural" sampling techniques a researcher might employ, and $1 + 5$ homogenization strategies, one unhomogenized and five homogenized corresponding to each salient variable with $\varepsilon = 0.025$. We then evaluated each model on a fresh evaluation set sampled from a mixture of the four unhomogenized distributions (T2T, DCFG, RCFG, BAL).

### 6.3 RESULTS

The original performances and improvements created by homogenizing different random variables can be found in Table 3. On average, homogenizing the DCFG and T2T distributions caused the accuracy to increase by 5.00pp and 2.84pp, respectively.

We note that the Calculator domain is much simpler than Karel when considering both input complexity (grid worlds versus arithmetic expressions) and output complexity (a DSL program versus a single digit). Furthermore, the difference in distributions between the naive sampling approaches and the versions with one homogenized random variable are not as different in Calculator as what we observed in Karel (see Table 1 for the dramatic effect of $\mathcal{D}_{marker}$). We hypothesize that it is this difference in complexity that explains the smaller (but still consistent) effect of homogenizing salient random variables in Calculator as compared to in Karel.

## 7 CONCLUSION

We demonstrate that existing sampling methods for randomly generating input-output examples have unintended and overlooked distribution flaws in both the Calculator and the Karel domain. These flaws prevent models trained on these distributions from generalizing to other test distributions, even if the are very simple. To resolve these problems, we propose a robust strategy for controlling and evaluating the bias of synthetic data distributions over programs and specifications by defining certain random variables that capture desired features of the program and input spaces, such as the number of parentheses in a calculator expression, and specifically manipulating their distributions. Equipped with our method, deep networks exhibit an increase in cross-distribution test accuracy, at the expense of a minor decrease in on-distribution test accuracy. We believe this methodology would lead to more rigorous evaluation of the synthesis techniques and moreover, aid them in learning better models that generalize well.

Equipped with a set of hand-designed salient random variables, we demonstrate the effectiveness of homogenizing synthetic datasets over this set. One of the core limitations of our approach is that the salient random variables are engineered by hand. This requires the scientist to have insights about the structure of the training examples they are randomly generating. Therefore, a promising extension of this algorithm is to automatically select which salient random variables to use, and automatically compute these variables; potentially via the use of a general unsupervised learning algorithm.

We evaluate our method on two domains: the Karel DSL, and a calculator expression parser. There is a natural question of whether the methods developed in this paper will improve out-of-distribution generalization on applications other than program synthesis which use synthetic data—for example, a convolutional neural network that receives renderings of a virtual environment, for the robotics and vision domains mentioned in Section 2.1. Providing a thorough evaluation of our proposed homogenization algorithm on alternative domains is a promising area for future work.

## ACKNOWLEDGEMENTS

This material is in part based upon work supported by the National Science Foundation under Grant No. TWC-1409915, Berkeley Deep Drive, and DARPA D3M under Grant No. FA8750-17-2-0091. Any opinions, findings, and conclusions or recommendations expressed in this material are those of the author(s) and do not necessarily reflect the views of the National Science Foundation and DARPA.

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

## A    THE KAREL DOMAIN

$$
\begin{aligned}
\text{Prog } p \quad &:= \quad \texttt{def main():}s \\
\text{Stmt } s \quad &:= \quad \texttt{while}(b):s \mid \texttt{repeat}(r):s \mid s_1;s_2 \\
&\quad \mid \quad a \mid \texttt{if}(b):s \mid \texttt{if}(b):s_1\texttt{else}:s_2 \\
\text{Cond } b \quad &:= \quad \texttt{markersPresent()} \mid \texttt{leftIsClear()} \\
&\quad \mid \quad \texttt{rightIsClear()} \mid \texttt{frontIsClear()} \\
&\quad \mid \quad \texttt{not}(b) \\
\text{Action } a \quad &:= \quad \texttt{move()} \mid \texttt{turnLeft()} \mid \texttt{turnRight()} \\
&\quad \mid \quad \texttt{pickMarker()} \mid \texttt{putMarker()} \\
\text{Cste } r \quad &:= \quad 0 \mid 1 \mid \cdots \mid 19
\end{aligned}
$$

```
GridWidth: m
GridHeight: n
Markers: {(i,j,k)_l}_l
Walls: {(i,j)_t}_t
KarelLoc: (i,j)
Orientation: d ∈ {N,S,E,W}
```
$2 \le m, n \le 16; i \le m;$
$j \le n; 1 \le k \le 9$

(a)                                                            (b)

Figure 4: (a) The DSL of Karel programs taken from Bunel et al. (2018), and (b) a declarative specification of the space of valid input worlds for Karel programs.

A declarative specification of the space of valid input worlds to Karel programs is shown in Figure 4(b). As with Bunel et al. (2018), we assume a bound on the input grid size to be $2 \le m, n \le 16$. Each cell $(i, j)$ in a grid can either be empty, contain an obstacle (i.e. a wall, specified by the list `Walls`), or contain $k \le 9$ markers (defined using the list `Markers`). The agent starts at some cell denoted by `KarelLoc` in the grid (which may contain markers but no obstacle) with a particular orientation direction denoted by `Orientation`.

The grammar for the Karel DSL we consider in this work is shown in Figure 4(a). The DSL allows the Karel agent to perform a `move` action to move one step in the grid in the direction of the orientation, actions `turnLeft` and `turnRight` to change its orientation direction, and actions `pickMarker` and `putMarker` to manipulate markers. The language contains `if`, `ifElse`, `while` constructs with conditionals {`front`, `left`, `right`}`IsClear`, `markersPresent`, and their negations. The `repeat` construct allows for a fixed number of repetitions. Note that the language does not contain any variables or auxiliary functions.

## B    SALIENT VARIABLE HOMOGENIZATION ALGORITHM

The following is a more formal description of the Algorithm described in Section 3, as well as proofs of correctness, and an investigation of the $\epsilon$ parameter's pratical effect.

### B.1    FULL PSEUDOCODE

Let $\mathbb{S}$ be some space that is sampled by some original distribution $q : \mathbb{S} \to [0, 1]$. Let $\mathbb{X}$ be the space of a salient variable which is calculated by $\nu : \mathbb{S} \to \mathbb{X}$; it is a finite set. Let $\varepsilon$ be our tolerance.

**procedure** SAMPLEDHOMOGENIZE($q : \mathbb{S} \to [0, 1], \nu : \mathbb{S} \to \mathbb{X}, \varepsilon \in \mathbb{R}, n \in \mathbb{N}$)

    $C \leftarrow \{x \to 0 : x \in \mathbb{X}\}$        ▷ Set up a dictionary of counts of each value of $\mathbb{X}$

    $t \leftarrow 0$        ▷ The total number of samples seen

    $\mathcal{D} \leftarrow \{\}$        ▷ $\mathcal{D}$ is originally an empty multiset

    **while** $|\mathcal{D}| < n$ **do**

        $s \leftarrow$ a sample from $q$

        $C[\nu(s)] \leftarrow C[\nu(s)] + 1$

        $t \leftarrow t + 1$

        $p_{\min} \leftarrow \frac{\min_{x \in \mathbb{X}} C[x]}{t}$

        $p_{curr} \leftarrow \frac{C[\nu(s)]}{t}$

        $g \leftarrow \frac{p_{\min} + \varepsilon}{p_{curr} + \varepsilon}$

        $h \leftarrow \begin{cases} 1 & \text{with probability } g \\ 0 & \text{with probability } 1 - g \end{cases}$

> **if** $h = 1$ **then**
>     $\mathcal{D} \leftarrow \mathcal{D} \cup \{s\}$
> **end if**
> **end while**
> **return** $\mathcal{D}$
> **end procedure**

## B.2 PROOF OF CORRECTNESS

Let the initial sampling distribution be $q$ and the resulting distribution be $r$. We use the notation $P_r[X = x]$ to refer to the probability that $X = x$ given that $S$ is sampled from the distribution $r$. $C[x]$ refers to the count for the salient variable value $x \in \mathbb{X}$ among past samples from $q$, as defined in SAMPLEDHOMOGENIZE.

**Theorem.** *The Salient Variable Homogenization algorithm produces samples from a distribution close to uniform. Formally, after $\frac{48 \log \frac{2|\mathbb{X}|}{\delta}}{p_m |\mathbb{X}|^2 \xi^2}$ samples are drawn from distribution $q$, we have that the resulting homogenized distribution satisfies*

$$\left| P_r[X = x] - \frac{1}{|\mathbb{X}|} \right| \le \xi$$

*with probability at least $1 - \delta$, where $p_m = \min_{x \in \mathbb{X}} P_q[X = x] > 0$ and $\varepsilon$ has been set to 0.*

*Proof.* We can simplify the probability as

$$P_r[X = x] = \frac{P_q[X = x]/C[x]}{\sum_{z \in \mathbb{X}} P_q[X = z]/C[z]}$$

(for $\varepsilon = 0$; see Probability Simplification Lemma below).

Let $\alpha = \frac{\xi |\mathbb{X}|}{4}$. We have that $n > \frac{3 \log \frac{2|\mathbb{X}|}{\delta}}{\alpha^2 p_m}$ by assumption and substituting in $\alpha$. By the Count Bounding Lemma we have that $\left| \frac{n P_q[X=x]}{C[x]} - 1 \right| \le \alpha$ for all $x$ with probability at least $1 - \delta$. The following computations assume $\left| \frac{n P_q[X=x]}{C[x]} - 1 \right| \le \alpha$ for all $x$ and thus are valid with probability $1 - \delta$.

We have $\frac{n P_q[X=x]}{C[x]} \in [1 - \alpha, 1 + \alpha]$. We also have

$$\left| \frac{1}{|\mathbb{X}|} \sum_{z \in \mathbb{X}} n P[X = z]/C[z] - 1 \right| = \left| \frac{1}{|\mathbb{X}|} \sum_{z \in \mathbb{X}} (n P[X = z]/C[z] - 1) \right|$$

$$=\le \frac{1}{|\mathbb{X}|} \sum_{z \in \mathbb{X}} |n P[X = z]/C[z] - 1|$$

$$=\le \frac{1}{|\mathbb{X}|} \sum_{z \in \mathbb{X}} \alpha$$

$$= \alpha$$

and thus $\frac{1}{|\mathbb{X}|} \sum_{z \in \mathbb{X}} \frac{n P[X=z]}{C[z]} \in [1 - \alpha, 1 + \alpha]$.

Combining the previous two ranges via a division, we have

$$|\mathbb{X}| P_r[X = x] = \frac{\frac{n P_q[X=x]}{C[x]}}{\frac{1}{|\mathbb{X}|} \sum_{z \in \mathbb{X}} \frac{n P[X=z]}{C[z]}} \in \left[ \frac{1 - \alpha}{1 + \alpha}, \frac{1 + \alpha}{1 - \alpha} \right]$$

We have that $\frac{1-\alpha}{1+\alpha} = 1 - \frac{2\alpha}{1+\alpha} \ge 1 - 2\alpha \ge 1 - 4\alpha$ and $\frac{1+\alpha}{1-\alpha} = 1 + \frac{2\alpha}{1-\alpha} = 1 + \frac{4\alpha}{2-2\alpha} \le 1 + 4\alpha$ since $2 - 2\alpha > 1$ for small $\alpha$. Thus, we have that $\left[ \frac{1-\alpha}{1+\alpha}, \frac{1+\alpha}{1-\alpha} \right] \subseteq [1 - 4\alpha, 1 + 4\alpha]$ and therefore we have

$$|\mathbb{X}|P_r[X = x] \in [1 - 4\alpha, 1 + 4\alpha]$$

we thus have that $\left|P_r[X = x] - \frac{1}{|\mathbb{X}|}\right| \leq \frac{4\alpha}{|\mathbb{X}|} = \xi$, completing our proof. $\qquad \square$

**Lemma** (Count Bounding). *We have that if $n > \frac{3 \log \frac{2|\mathbb{X}|}{\delta}}{\alpha^2 p_m}$ samples have been drawn from $q$ that*

$$P\left[\exists x \in \mathbb{X}, \left|\frac{nP_q[X = x]}{C[x]} - 1\right| \geq \alpha\right] \leq \delta$$

*where $p_m = \min_{x \in \mathbb{X}} P_q[X = x]$*

*Proof.* We can model $C[x]$ as a sum of $n$ independent Bernoulli trials with probability $P_q[X = x]$. Using Chernoff bound, we have that

$$P[C[x] \geq nP_q[X = x](1 + \alpha)] \leq e^{-\alpha^2 nP_q[X=x]/3}$$

and

$$P[C[x] \geq nP_q[X = x](1 - \sqrt{2/3}\alpha)] \leq e^{-\alpha^2 nP_q[X=x]/3}$$

Thus, we have by union bound that

$$P\left[\frac{nP_q[X = x]}{C[x]} \geq \frac{1}{1 - \sqrt{2/3}\alpha} \vee \frac{nP_q[X = x]}{C[x]} \leq \frac{1}{1 + \alpha}\right] \leq 2e^{-\alpha^2 nP_q[X=x]/3}$$

For $\alpha \leq \sqrt{3/2} - 1$ we have that $\frac{1}{1-\sqrt{2/3}\alpha} = 1 + \frac{\sqrt{2/3}\alpha}{1-\sqrt{2/3}\alpha} = 1 + \frac{\alpha}{\sqrt{3/2}-\alpha} \leq 1 + \alpha$ and $\frac{1}{1+\alpha} = 1 - \frac{\alpha}{1+\alpha} \geq 1 - \alpha$. Thus, we can restate the previous inequality as

$$P\left[\frac{nP_q[X = x]}{C[x]} \geq 1 + \alpha \vee \frac{nP_q[X = x]}{C[x]} \leq 1 - \alpha\right] \leq 2e^{-\alpha^2 nP_q[X=x]/3}$$

or in other words

$$P\left[\left|\frac{nP_q[X = x]}{C[x]} - 1\right| \geq \alpha\right] \leq 2e^{-\alpha^2 nP_q[X=x]/3}$$

Thus we can bound the RHS as

$$2e^{-\alpha^2 nP_q[X=x]/3} \leq 2e^{-\alpha^2 np_m/3} < 2e^{-\alpha^2 \frac{3 \log \frac{2|\mathbb{X}|}{\delta}}{\alpha^2 p_m} p_m/3} = 2e^{-\log \frac{2|\mathbb{X}|}{\delta}} = \frac{\delta}{|\mathbb{X}|}$$

where the first inequality is since $P_q[X = x] \geq p_m$ and the second is since $n > \frac{3 \log \frac{2|\mathbb{X}|}{\delta}}{\alpha^2 p_m}$.

We can again apply union bound to get that

$$P\left[\exists x \in \mathbb{X}, \left|\frac{nP_q[X = x]}{C[x]} - 1\right| \geq \alpha\right] \leq |\mathbb{X}|\frac{\delta}{\mathbb{X}} = \delta$$

$\qquad \square$

**Lemma** (Probability Simplification).

$$P_r[X = x] = \frac{P_q[X = x]/C[x]}{\sum_{z \in \mathbb{X}} P_q[X = z]/C[z]}$$

*Proof.*

$$P_r[X = x] = \sum_{s \in \mathbb{S}: \nu(s)=x} P_r[S = s]$$

$$= \sum_{s \in \mathbb{S}: \nu(s)=x} \frac{g(s)q(s)}{\sum_{s' \in \mathbb{S}} g(s')q(s')}$$

$$= \frac{\sum_{s \in \mathbb{S}: \nu(s)=x} g(s)q(s)}{\sum_{s' \in \mathbb{S}} g(s')q(s')}$$

$$= \frac{\sum_{s \in \mathbb{S}: \nu(s)=x} g(s)q(s)}{\sum_{z \in \mathbb{X}} \sum_{s' \in \mathbb{S}: \nu(s')=z} g(s')q(s')}$$

Since we can simplify

$$\sum_{s \in \mathbb{S}: \nu(s)=x} g(s)q(s) = \sum_{s \in \mathbb{S}: \nu(s)=x} \frac{\min_{y \in \mathbb{X}} C[y]}{C[\nu(s)]} q(s)$$

$$= \frac{\min_{y \in \mathbb{X}} C[y]}{C[x]} \sum_{s \in \mathbb{S}: \nu(s)=x} q(s)$$

$$= \frac{\min_{y \in \mathbb{X}} C[y]}{C[x]} P_q[X = x]$$

we have that

$$P_r[X = x] = \frac{\sum_{s \in \mathbb{S}: \nu(s)=x} g(s)q(s)}{\sum_{y \in \mathbb{X}} \sum_{s' \in \mathbb{S}: \nu(s')=y} g(s')q(s')}$$

$$= \frac{\frac{\min_{y \in \mathbb{X}} C[y]}{C[x]} P_q[X = x]}{\sum_{z \in \mathbb{X}} \frac{\min_{y \in \mathbb{X}} C[y]}{C[z]} P_q[X = z]}$$

$$= \frac{P_q[X = x]/C[x]}{\sum_{z \in \mathbb{X}} P_q[X = z]/C[z]}$$

$\square$

## B.3 Efficiency Analysis

We show that the number of samples from the original distribution required to produce a sample from the homogenized distribution is $O(\frac{1}{\varepsilon})$ in expectation.

In the Salient Variable Homogenization algorithm, we have the probability of not rejecting a given sample as $g(s) = \frac{\min_x P[X=x]+\varepsilon}{P[X=X(s)]+\varepsilon}$. We know that

$$g(s) = \frac{\min_x P[X = x] + \varepsilon}{P[X = v(s)] + \varepsilon} \geq \frac{\varepsilon}{P[X = v(s)] + \varepsilon} \geq \frac{\varepsilon}{1 + \varepsilon}$$

Since each sample is independent, we can model this as a geometric distribution, and thus we have that the expected number of tries $t = \frac{1}{g(s)} \leq 1 + \frac{1}{\varepsilon}$. We thus have that in expectation, we need to sample $O(\frac{1}{\varepsilon})$ samples from the original distribution to produce one homogenized sample.

## B.4 Empirical Effect Of Varying $\varepsilon$

The solid line is an upper bound $\frac{\varepsilon}{1+\varepsilon}$ derived in Section B.3. Seen in the measured samples for different values of $\varepsilon$, the upper bound appropriately reflects the maximum height of the samples, with

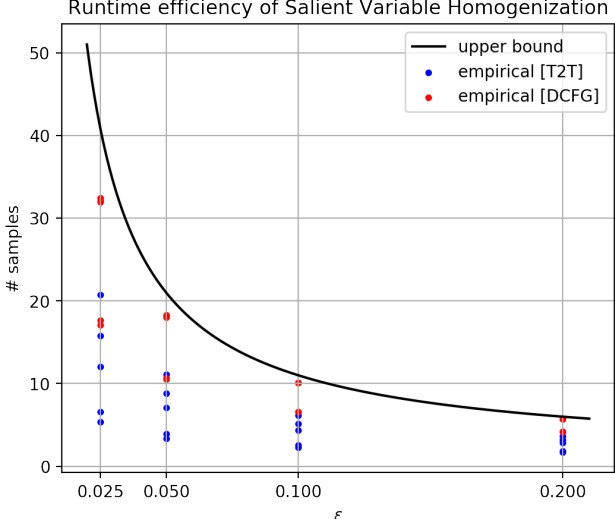

Figure 5: Number of samples required by salient variable homogenization parameterized by an $\varepsilon$ before a new sample is returned.

| | Original | $\varepsilon = 0.025$ | $\varepsilon = 0.050$ | $\varepsilon = 0.100$ | $\varepsilon = 0.200$ |
|---|---|---|---|---|---|
| T2T | 83.83% | +2.84pp | +1.31pp | +1.33pp | +2.45pp |
| DCFG | 78.25% | +5.00pp | +4.50pp | +3.54pp | +3.42pp |

Table 4: Improvements in Calculator performance with homogenized datasets of various sampling parameters $\varepsilon$. See Section 6 for details on performance metrics.

very little remaining space on the DCFG dataset and some but not much on the T2T dataset, and is thus a close bound. As the values of $\varepsilon \to \infty$ the bound approaches the limit 1, indicating no samples are rejected by the algorithm.

Increasing the parameter $\varepsilon$ has the theoretical affect of causing the homogenized distribution to deviate more from uniform in its salient random variables, as shown in B.3. In practice, we discover that performance boosts tend to decrease with increasing $\varepsilon$, although the effect was not as pronounced on the T2T dataset, potentially because the unhomogenized T2T distribution is closer to uniform and thus homogenization has a limited effect for any larger $\varepsilon$ values.

### B.5 EMPIRICAL EVIDENCE FOR INCREASE IN UNIFORMITY

Empirically, the Salient Variable Homogenization algorithm led to increases in uniformity in the variable being homogenized. We measure uniformity by KL divergence between the distribution being measured and the uniform distribution. A table of relative improvements is given in Table 5.

| | Length | Max Depth | Mean Depth | #Operations | #Parens |
|---|---|---|---|---|---|
| DCFG | 42.98% | 30.77% | 27.05% | 43.95% | 23.63% |
| T2T | 46.68% | 30.45% | 13.99% | 38.82% | 36.91% |

Table 5: Percentage reductions in KL-divergence from uniform of the given salient variable when homogenized at $\varepsilon = 0.025$.

