# OpenReview forum: "Synthetic Datasets for Neural Program Synthesis"
_ICLR.cc/2019/Conference_

### Official Review · AnonReviewer3 · 2018-11-02
**Lacking arguments why the proposed method generalizes well to other problem settings**

**Rating:** 6
**Confidence:** 2

**Review:**

The paper presents a methodology for improved program synthesis by generating datasets for program induction and synthetic tasks from uniform distributions. This method is evaluate on two problem settings.

The methodology is presented in section 3. Even though the outline does not seem to be complicated, the presentation in section 3 leaves me puzzled. The the second paragraph two sets of silent variables are introduced X_1,...,X_n and Z_1,...,Z_m but never used again the rest of the paper. In the third and forth paragraph details about the Karel domain are presented without the Karel domain having been introduced. It seems you are using rejection sampling to sample from a uniform distribution. Why can you not sample from a uniform distribution directly? What do you mean with the notation X(s)? What are you proving in Appendix? Would maybe be clearer if you presented it as a theorem/lemma.

The remaining part of the paper evaluates this methodology on two specific problem settings, the Karel domain and Calculator domain. The generalization performance is increased when trained on datasets generated by the method presented in the paper. However, I cannot find and strong arguments in the paper why this property should generalize to other problem settings. To me the analysis and experimental results seems to be tailored to the two problems settings used in the paper.

==== After revision ====

The authors have done a great job addressing the concerns I had about the clarity. Consequently, I have raised my score, whereas my fairly low confidence still remains.

---

> ### Author Response · Authors · 2018-11-25
> **Response to review**
>
> We would like to thank the reviewer for their helpful and insightful comments. Our responses to the specific concerns follow.
>
> > The the second paragraph two sets of silent variables are introduced X_1,...,X_n and Z_1,...,Z_m but never used again the rest of the paper.
> We have removed them from the paper.
>
> > In the third and forth paragraph details about the Karel domain are presented without the Karel domain having been introduced.
> We intend Section 3 to be a general description of the issues facing synthetic data generation that is independent of any given domain; however, to make the exposition clearer and more concrete, we used some examples from the Karel domain for illustration, for those who are already familiar with the domain from the related work.
>
> > It seems you are using rejection sampling to sample from a uniform distribution. Why can you not sample from a uniform distribution directly?
> We are attempting to make certain features (salient variables) uniform over the distribution of examples, in a way that cannot be easily handled generatively. For example, sampling Calculator expressions with uniform length requires a much more complicated algorithm than any of the standard algorithms for sampling programs from a CFG. Furthermore, the domain may place complicated restrictions on which subset of examples are valid. Our method allows for the usage of an underlying arbitrary sampling method, as long as all values of a given salient variable are sufficiently represented.
>
> > What do you mean with the notation X(s)?
> It means “the value of the salient variable X within sample s”. However, we have replaced this notation in our new revision.
>
> > What is you proving in Appendix? Would maybe be clearer if you presented it as a theorem/lemma.
> In the appendix, we now show a probabilistic bound on the uniformity of the salient variable in the resulting distribution. We present it as a series of theorems and lemmas.
>
> We have revamped the proof in the appendix to account for the fact that we use empirical estimates of the distribution over the salient variable X, rather than the true distribution. We now provide a full description of the algorithm as pseudocode, and have updated the notation throughout the description and the proof for further clarity. While the proof is now entirely new, we have nevertheless tried to ameliorate any past issues regarding notation.
>
> > However, I cannot find and strong arguments in the paper why this property should generalize to other problem settings. To me the analysis and experimental results seems to be tailored to the two problems settings used in the paper.
> We believe that the property should generalize to other problem settings for the following reasons:
>
> 1. Other domains/methods such as RobustFill [1] also use randomly-generated synthetic training data; the authors state in Section 3.3 that “Intuitively, it is possible to generalize [...] using randomly synthesized training because the model is learning function semantics, rather than a particular data distribution.”
> Nevertheless, the method’s reported performance on a manually-curated test set (Figure 4) is significantly lower than on a synthetic validation set drawn from the same distribution as the training data, which shows that the model adapts significantly adapts to the particulars of the training distribution.
> Given that this method exhibits similar failures given a similarly generated synthetic training dataset, we expect it to also improve similarly given more carefully generated training data.
>
> 2. Our Karel and calculator domains have little in common in terms of semantics; one is for inferring a program which controls an agent’s movements within a gridworld, whereas the other is learning to perform arithmetic. Nevertheless, the same method of defining salient random variables and making them more uniform throughout the training data increased generalization performance for both domains.
>
> 3. Even though these domains have highly different semantics, they (and indeed, most other program synthesis tasks which involve trees) share some salient random variables such as length, depth of nesting, and number of operations.
>
> 4. On the calculator domain, all salient variables that we thought of had a positive effect on eventual accuracy, suggesting that the result is not too brittle with respect to the choice of salient variable (though some work better than others) and therefore similar results will be easier to obtain on other domains.
>
> [1] RobustFill: Neural Program Learning under Noisy I/O. Jacob Devlin, Jonathan Uesato, Surya Bhupatiraju, Rishabh Singh, Abdel-rahman Mohamed, Pushmeet Kohli. https://arxiv.org/abs/1703.07469
>
> > To achieve good generalization performance the important To me it seems that the important
> We weren’t sure about this point in the review; can we answer a question that got missed in a copy and paste issue or similar?

---

> > ### Comment · AnonReviewer3 · 2018-12-03
> > **Clarity improved**
> >
> > The authors have done a great job addressing the concerns I had about the clarity. Consequently, I have raised my score, whereas my fairly low confidence still remains.

---

### Official Review · AnonReviewer1 · 2018-11-03
**Nice presentation of a serious issue, with some flaws**

**Rating:** 6
**Confidence:** 4

**Review:**

This paper provides a good presentation of a serious problem in evaluating (as well as training!) performance of machine learning models for program synthesis / program induction: considering specifically the problem of learning a program which corresponds to given input/output pairs, since large datasets of "real-world" programs typically do not exist it necessary to construct a synthetic dataset for training and testing; this requires both (a) generating programs, and (b) generating input/output examples for these programs. Enumerating either all possible programs or examples is typically impossible, and so a sampling scheme is used to simulate "reasonable" programs and examples. This may hinder generalization to other data not often produced by the sampling scheme.

To address this, the paper then argues that programs should be synthesized from a distribution which as as uniform as possible over a set of user-specified statistics (the "salient variables") as well as over the input space. Intuitively, this makes sense: maximizing the entropy of the synthetic data should provide good coverage over the entire input space. However, there are a few ways in which the particular approach is unsatisfying:

(1) It requires manual curation of salient random variables. This sort of punts the decision of "what should my sampling procedure be" to "what is my choice of salient variables to make uniform". I agree that this is still an improvement.

(2) The procedure described for generating synthetic examples is essentially a rejection sampling algorithm, and it will fail to generate examples in a reasonable timeframe if the original proposal distribution is highly non-uniform, or if the salient random variables include values which fall in the tail of the proposal distribution.

Also, relatedly, I don't follow the description of correctness in section 8.2 at all. What is meant by the "= 1" at the end of the line right before "… And thus…"? Clearly P_r[X=x] cannot both equal 1, and equal k. Is the "=1" meant to only mean the summand itself? If so, please fix the notation. Also, I assume that k is meant to be the cardinality of the set {s: X(s) = x}, but this is not defined anywhere. Notational issues aside, unless the mapping X(s) from sample to salient variable is one-to-one, then I'm not clear how the P_q[X = X(s)] would relate to q(s). This should be made more clear. Also, I believe there need to be conditions on q(s), e.g. such that min_x P_q[X = x] must always be greater than zero.


These issues aside, the empirical demonstrations on the Karel the Robot examples are nicely presented and make the point well. My primary question here would be around section 5, the "real-world benchmarks", where it is observed that the baseline model performs less well than re-training on a uniform / homogenized dataset. While it is nice that it performed better, I don't understand why even the better number (19.4%) is so low; the performance of the uniform model in table 1 tends to be much higher (in the 60% to 70% range). This would suggest that the uniform model perhaps is significantly *underweighting* important parts of the space. What is causing this? e.g. what do the salient variables look like for real-world examples?


Finally, I am not sure I understand how the calculator example fits into this paper. Unless I misunderstand, it is not a program synthesis task, but rather a regression task. Clearly it does still depend on generation of synthetic data, but that is more a different task (as described in section 2). I feel its inclusion somewhat dilutes the paper. Rather, it would be nice to see more discussion or investigation into the failure modes of these trained models; for example, looking deeper at the handling of control flow and recursion, or at whether particular values of salient variables tended to be correlated with success or failure under different train / test regimes.



===== after updates =====

Thanks for the edits — I believe the overall paper is more clearly presented, now.

I still think it is a stretch to consider the calculator domain is a program induction problem: it is a regression problem, from an input string to an output integer, or alternately a classification problem, since it computes the result mod 10. The only way I could understand this as a program induction problem is rather obliquely, if the meaning is that any system which is able to compute the result of the calculator evaluation has implicitly replicated internally, in some capacity, the sequence of instructions which are evaluated. I don't think this is really very clear though; for example, given two calculator programs, one a subprogram of another (e.g., "4*(3+2)" and "3+2"), do the resulting "induced" computations share the same compositional structure? The examples of program induction in section 2 are largely architectures which are explicitly designed to have properties which mimic conventional programming languages (e.g. extra data structures as memories, compositionality, …). In contrast, the calculator example in this paper simply uses an LSTM.

That said, I think it's still a great example! Learning a fast differentiable model which accurately mimics existing non-differentiable model has tons of applications, and has exactly the same challenges regarding synthetic data.



I have to say I find the new section 8.3 a bit intuitively challenging; e.g. it's not clear really how long a waiting time of 48 log(2|X|/\delta) / (p|X|^2 z^2) really is. But, to that end, I appreciate the empirical discussion in 8.4–8.6.

I've updated my review to increase my score — I lean towards accepting this paper, as it is a timely contribution and I think it is important for future program synthesis papers to take the results here to heart. I've reduced my confidence slightly, as I have not fully reviewed the new proof in 8.3.

---

> ### Author Response · Authors · 2018-11-25
> **Response to review**
>
> We would like to thank the reviewer for their helpful and insightful comments. Our responses to the specific concerns follow.
>
> > (1) It requires manual curation of salient random variables. This sort of punts the decision of "what should my sampling procedure be" to "what is my choice of salient variables to make uniform". I agree that this is still an improvement.
> A key contribution of our paper is the identification of issues with training and evaluation datasets of current neural program synthesis approaches, and a first step towards alleviating them using salient random variables. As the reviewer states, this is still an improvement over the status quo of randomly sampling programs from a DSL, and we leave the automatic discovery and curation of salient random variables to future work. Additionally, we found that all salient variables that we tried on the Calculator task had a positive effect on eventual accuracy, suggesting that the exact choice of salient variables is not significant to improve the overall results, though some work better than others.
>
> > (2) The procedure described for generating synthetic examples is essentially a rejection sampling algorithm [...]
> As stated in Section 3, the need for training examples to satisfy complex constraints in more complicated domains like Karel makes it very difficult to use other methods for generating random examples, while ensuring that a salient variable follows a particular distribution. Furthermore, a rejection sampling approach is easy to graft onto existing sampling approaches as we have done in this paper. Nevertheless, we can reduce the runtime needed for the procedure described in Section 3 by adjusting the epsilon hyperparameter (at the cost of decreasing the uniformity of the result).
>
> > Also, relatedly, I don't follow the description of correctness in section 8.2 at all
> We have significantly revised the description of the method in Section 3, and revamped the proof in Section 8.2 (now Section 8.3) to account for the fact that we use empirical estimates of the distribution over the salient variable X, rather than the true distribution. We now provide a full description of the algorithm as pseudocode, and have updated the notation throughout the description and the proof for further clarity. While the proof is now entirely new, we have nevertheless tried to ameliorate any past issues regarding notation.
>
> > Also, I believe there need to be conditions on q(s), e.g. such that min_x P_q[X = x] [...].
> We now state this condition in the proof.
>
> > I don't understand why even the better number (19.4%) is so low; the performance of the uniform model in table 1 tends to be much higher (in the 60% to 70% range). This would suggest that the uniform model perhaps is significantly *underweighting* important parts of the space. What is causing this?
> This seems to be the main point of confusion in the review and we have also addressed in the paper revision. While the model’s performance grows significantly from 11.1% to 19.4% on real-world tasks, it is important to note that it is not because the uniform model is “underweighting” important parts of the search space that is causing the lower overall accuracy; but rather the following two challenges play a more important role on these problems:
> - Many of the real-world problems require long programs to solve, which are intrinsically difficult for the model to synthesize correctly even if trained with more uniform data. For example, 86.11% of the programs in the real-world test set contain more than 10 tokens, whereas 75.56% of the synthetic test set does.
> - In the real-world examples, the specification always contains fewer than 5 I/O pairs; indeed,  many only contain 1 I/O pair. However, the training methodology for the model assumes that it is provided with a diverse set of 5 I/O pairs.
>
> > Finally, I am not sure I understand how the calculator example fits into this paper. Unless I misunderstand, it is not a program synthesis task, but rather a regression task. Clearly it does still depend on generation of synthetic data [...]
> We included the calculator example to show that both the notion of salient variables and making them more uniform while generating synthetic data generalize to different tasks/domains. The calculator example is not a program synthesis task, in that the model’s output is not a program; however, we would classify it under program induction, which is a closely related task and is often trained using synthetically generated data (for example, in the Learning to Execute paper [1] where one of the tasks was computing the sum of two numbers). In particular, both the Karel programs and the calculator expressions are generated from trees drawn from a context-free grammar.
>
> Please let us know if this helped clarify the questions and concerns, and let us know if there are any more questions.
>
> [1] Learning to Execute. Wojciech Zaremba, Ilya Sutskever. https://arxiv.org/abs/1410.4615

---

### Official Review · AnonReviewer4 · 2018-11-13
**Nice evaluations, empirically sound methodology, but no new model**

**Rating:** 7
**Confidence:** 3

**Review:**

This is a nice paper. It makes novel contributions by investigating (a) the problem of skewed dataset distributions in neural program synthesis, specifically program induction from given I/O pairs, and (b) the extent to which making them uniform would improve model performance.

The paper argues that there are inevitable and artificial sparsities as well as skews in existing datasets (e.g. pruning illegal I/O pairs, naive random sampling tends not to generate complex nested control-flow statements), and the principled way to minimize these sparsities and skews is to make distributions over salient random variables uniform. The authors evaluate their hypothesis empirically on two flavors of neural program synthesis methods: program inductions on explicit DSL represented by Karel, and implicit differentiable neural program synthesizers (such as stack, RAM, GPU as cited in section 2) represented by a Calculator example. In evaluations, they construct few challenging “narrower” datasets and show baseline models perform significantly worse than models trained on datasets with uniform distributions (by 39-66 pp). Along this line, they also show uniform models consistently perform much better than baseline ones on other out-of-distribution test sets. To show how bad a model would perform if it were trained on a skewed training set, they train models on narrower datasets and evaluate them on different narrower sets.

The strength of this paper are:
(1) It has complete and empirically sound evaluations: both showing how much better uniform models would be and how much worse non-uniform models would be.

(2) Although we might doubt the salient random variables are handcrafted and rejection sampling wouldn’t make the dataset completely uniform, they include evaluations on out-of-distribution datasets (e.g. CS106A dataset in section 5.2) to show that uniform models still perform better and thus their sampling scheme does cover some non-obvious sparsities and skews.

(3) Despite the doubt on efficiencies of rejection sampling, they include both a proof and empirical results (section 8.3 and 8.4) to show they need sample O(1/ε) times before finishing.

Weaknesses:
(1) No new model. This work has solely using the existing model from Bunel et al. (2018) in the Karel domain and didn’t propose a new model that illustrates possibly a way to utilize/demonstrate the uniformity of dataset.

(2) The calculator example is relatively too trivial to represent the whole genre of implicit differentiable neural program synthesizer (e.g. stack, GPU, RAM).

(3) No statistical tests (such as chi-square test) to support the claim about uniformity (even on chosen salient variables)

Questions:
(1) What if the distribution of real-world programs are skewed and neural synthesizers are supposed to take advantage of their skewness?

(2) Why would you claim the calculator example is not a program synthesis task while intending to use it to represent another genre of program synthesis methods?

Suggestions:
(1) To show that current salient random variables do not make the dataset theoretically uniform but are still approximate enough, why not construct some distinct held-out salient variables (such as memory/grid/marker query times, executing time) from existing ones, construct narrower test sets accordingly, and hopefully show uniform models still perform significantly better than baseline?

(2) In section 8.2, why not write the proportionality statement in two lines so that people wouldn’t be confused to think Pr[X=x] = 1 while intending to show Pr[X=x] ∝ 1(an arbitrary constant) so that Pr[X] is uniform?

---

> ### Author Response · Authors · 2018-11-25
> **Response to review**
>
> We would like to thank the reviewer for their helpful and insightful comments. Our responses to the specific concerns follow.
>
> > (1) No new model
> In this paper, we chose to focus on the impact of dataset generation and the training process on the performance of existing program synthesis models, which has largely been ignored in these neural synthesis works. We leave the impact of changes to the model for future work.
>
> > (2) The calculator example is relatively too trivial to represent the whole genre of implicit differentiable neural program synthesizer
> While the calculator example is indeed fairly simple, we did not intend it to be a representative of all differentiable neural program synthesizers. Rather, we wanted to show that the results generalize to domains other than Karel. Furthermore, due to its relative simplicity, we were able to perform a wide variety of systematic experiments as reported in the paper.
>
> > (3) No statistical tests (such as chi-square test) to support the claim about uniformity (even on chosen salient variables)
> We have added a section to the appendix where we report the KL divergence between the uniform distribution and the generated data’s empirical distribution for the Calculator domain.
>
> > (1) What if the distribution of real-world programs are skewed and neural synthesizers are supposed to take advantage of their skewness?
> We fully agree that it would be ideal for neural synthesizers to take advantage of any skew present in the distribution of real-world programs, or more generally, the distribution of tasks that real users are want to solve; indeed, especially for program synthesis tasks from input-output examples, there may exist a large number of spurious programs that satisfy the constraints but not the user’s intent.
>
> However, as it is difficult and expensive to construct such real-world datasets, learning from synthetic datasets is still useful and the common paradigm currently taken in the research literature. In this paper, we want to point out problems with existing ways that the synthetic datasets are constructed, and suggest improvements to mitigate some of these problems.
>
> > (2) Why would you claim the calculator example is not a program synthesis task while intending to use it to represent another genre of program synthesis methods?
> As explained in the first paragraph of the introduction, we make a distinction between “program synthesis” (where the model outputs programs in a DSL) and “program induction” (where we train a differentiable model end-to-end to represent the behavior of a program). We consider the calculator example to be an example of the latter, which is why we stated it’s not a program synthesis task. We have updated the paper to clarify this point.
>
> > (1) To show that current salient random variables do not make the dataset theoretically uniform but are still approximate enough [...]
> For the Calculator domain, we show that making only one of the salient variables uniform still improves performance across multiple distributions (including distributions unrelated to the one used to generate the training data).
>
> > (2) In section 8.2 [...]
> We have revamped the proof in Section 8.2 (now in Section 8.3) to account for the fact that we use empirical estimates of the distribution over the salient variable X, rather than the true distribution. While the proof is now entirely new, we have nevertheless tried to ameliorate any past issues regarding notation.
>
> Please let us know if this helped clarify the questions and concerns, and let us know if there are any more questions.

---

### Author Response · Authors · 2018-11-25
**Summary of revised paper**

Thanks to the reviewers for all the helpful suggestions and comments. We have uploaded a new paper revision with the following key changes:
1. Updated the notation in Section 3 (“Our Data Generation Methodology”) for further clarity
2. Clarified the challenges in real-world Karel tasks
3. Reported the result of the Action-Only Augmented model on the original test set (decrease in accuracy by less than 2 percentage points)
4. Stated that the Calculator domain is a program induction task
5. Detailed the procedure of Section 3 in pseudocode in the appendix, Section 8.2
6. Revamped the proof in the appendix, Section 8.3 to account for the fact that we use empirical estimates of the distribution.
7. Reported the empirical uniformity of salient variables on the Calculator datasets after applying the procedure from Section 3

---

### Meta-Review · Area_Chair1 · 2018-12-18
**Important observation, backed by solid work**

**Confidence:** 4
**Recommendation:** Accept (Poster)

**Metareview:**

This paper analyzes existing approaches to program induction from I/O pairs, and demonstrates that naively generating  I/O pairs results in a non-uniform sampling of salient variables, leading to poor performance. The paper convincingly shows, via strong evaluation, that uniform sampling of these variables can much result in much better models, both for explicit DSL and implicit, neural models. The reviewers feel the observation is an important one, and the paper does a good job providing sufficiently convincing evidence for it.

The reviewers and AC note the following potential weaknesses: (1) the paper does not propose a new model, but instead a different data generation strategy, somewhat limiting the novelty, (2) Salient variables that need to be uniformly sampled are still user specified, (3) there were a number of notation and clarity issues that make it difficult to understand the details of the approach, and finally, (4) there are concerns with the use of rejection sampling.

The authors provided major revisions that address the clarity issues, including an addition of new proofs, cleaner notation, and removal of unnecessary text. The authors also included additional results, such as KL divergence evaluation to show how uniform the distribution is. The authors also described the need for rejection sampling, especially for Karel dataset, and clarified why the Calculator domain, even though is not "program synthesis", still faces similar challenges. The reviewers agreed that not having a new model is not a chief concern, and that using rejection sampling is a reasonable first step, with more efficient techniques left for others for future work.

Overall, the reviewers agreed that the paper should be accepted. As reviewer 1 said it best, this paper "is a timely contribution and I think it is important for future program synthesis papers to take the results and message here to heart".